# "The phone number tells us good things we didn't know before." Use of interactive voice response calling for improving knowledge and uptake of family planning methods among Maasai in Tanzania

Kennedy Ngowi[1,2☯]*, Perry Msoka[1,2,3☯], Benson Mtesha[1☯], Jacqueline Kwayu[1,4], Tauta Mappi[1], Krisanta Kiwango[1], Ester Kiwelu[1], Titus Mmasi[5], Aifello Sichalwe[6], Benjamin C. Shayo[4,7], Eusebious Maro[4], I. Marion Sumari-de Boer[1,2,8,9]

1 Kilimanjaro Clinical Research Institute, Moshi, United Republic of Tanzania, 2 Amsterdam Institute for Global Health and Development, Amsterdam, The Netherlands, 3 Amsterdam Institute for Social Science Research (AISSR), Amsterdam, The Netherlands, 4 Kilimanjaro Christian Medical Center, Moshi, United Republic of Tanzania, 5 The office of District Medical office, Monduli, Arusha, United Republic of Tanzania, 6 The office of Regional Medical office, Arusha, United Republic of Tanzania, 7 Baylor College of Medicine, Houston, Texas, United States of America, 8 Kilimanjaro Christian Medical University College, Moshi, United Republic of Tanzania, 9 Knowledge, Technology and Innovation Group, Wageningen University & Research, Wageningen, the Netherlands

☯ These authors contributed equally to this work.
* k.ngowi@kcri.ac.tz

**Data Availability Statement:** Data are publicly available consisting of the survey, medical extraction, as well as IVCR system, are located at

## Abstract
## Introduction

Maasai living in the Arusha region, Tanzania, face challenges in feeding their children because of decreasing grazing grounds for their cattle. Therefore, they requested birth control methods. Previous studies have shown that lack of knowledge about, and poor access to, family planning (FP) may worsen the situation. We developed an interactive voice response calling (IVRC) platform for Maasai and health care workers (HCW) to create a venue for communication about FP to increase knowledge and access to FP. The objective of this study was to explore the effect of the platform on knowledge, access and use of family planning methods. We applied a participatory action research approach using mixed methods for data collection to develop and pilot-test an mHealth-platform with IVRC using Maa language. We enrolled Maasai-couples and HCW in Monduli District (Esilalei ward), Arusha Region, and followed them for 20 months. A baseline assessment was done to explore knowledge about FP. Furthermore, we abstracted information on FP clinic visits. Based on that, we developed a system called Embiotishu. A toll-free number was provided to interact with the system by calling with their phone. The system offers pre-recorded voice messages with information about FP and reproductive health to educate Maasai. The system recorded the number of calls and the type of information accessed. We measured the outcome by (1) a survey investigating the knowledge of contraceptive methods before and after Embiotishu and (2) counting the number of clinic visits (2018–2020) from medical records and feedback

https://www.ebi.ac.uk/biostudies/studies/S-BSST980#.

**Funding:** This work was supported by Voice.global grant no. A-05253-02-507496 under the linking and learning grant for Tanzania. The funders had no role in study design, data collection and analysis, decision to publish, or preparation of the manuscript.

**Competing interests:** The authors have declared that no competing interests exist.

from qualitative data for FP used among Maasai. The acceptability and feasibility were explored through focus group discussions (FGDs) with Maasai and in-depth interviews (IDIs) with HCW. We recruited 76 Maasai couples whom we interviewed during the baseline assessment. The overall knowledge of contraceptives increased significantly (p<0.005) in both men and women. The number of clinic visits rose from 137 in 2018 to 344 in 2019 and 228 in the first six months of 2020. Implants were the most prescribed family planning method, followed by injections and pills, as found in medical records. The number of incoming calls, missed calls, and questions were 24,033 over 20 months. Out of these calls, 14,547 topics were selected. The most selected topics were modern contraceptives (mainly implants, condoms, tubal ligation, and vasectomy). Natural methods of contraception (vaginal fluid observations, calendar, and temperature). Our study has shown that the IVRC system led to an improvement in knowledge about and access to contraceptives. Furthermore, it has potential to increase access to health information as well as improve dialogue between Health workers and Maasai.

## Introduction

Tanzania has a relatively high fertility rate, with about 6.97 children per mother, according to the World Health Organization (WHO) [1]. In comparison, globally, the rate was 3.0 in 2013 [2]. Limited data is available on fertility rates among Maasai. Still, data from the year 2000 showed that in Tanzania, Maasai women in the age category 45–49 have a fertility rate of more than 7, and the mean number of children among married women was 6.4 [3]. In 2016, of contraceptives' use and unmet need for family planning among married women in Tanzania was 38.4% and 22%, respectively [4]. Despite the scarcity of published data, some studies indicate that Maasai are unwilling to use family planning methods as they highly value procreation and wish to have many children. As a result, the use of contraceptives is low, and there is a high rate of unintended pregnancies [5–7]. Consequently, the pastoral community faces a dire challenge to provide for their children and families as grazing space has become increasingly limited.

Tanzania has many health services, including reproductive health and family planning services. However, accessibility is low in remote areas, especially in marginalized communities like the Maasai. Furthermore, Maasai may lack adequate knowledge about family planning and other reproductive health issues. For Maasai women, becoming a mother is a sign of procreation [5]. Several sexual practices are related, such as early sexual debut, female circumcision, polygamy and traditional birth outside the hospital [8]. Poor uptake of reproductive services and contraceptives among women living in rural areas in Tanzania was mentioned to be associated with sociocultural factors [9]. A study among young Maasai women in Kenya showed that the rate of family planning use is low, while there is a high rate of unintended pregnancies [7]. Further, they also found that women think family planning is only used by married women who do not want more children and that family planning methods cause side effects such as infertility and cancer.

Although, mobile phone coverage in Tanzania is extremely high, with >80%, and several mHealth programs have been deployed for family planning and reproductive health, the accessibility of such programs poses a challenge to many Maasai who are illiterate and only speak Maa, which is their tribal language [10]. These programs, such as 'Wazazi Nipendeni' and 'Mobile for Reproductive Health', are in the Kiswahili language and use SMS (Short Message Service) or text messages [11–13].

Several recommendations have been made, including using Interactive voice response calling (IVRC) as a potential application to improve communication and increase knowledge on health-related issues among the low-literacy community [14]. Interactive voice response calling (IVRC) uses a toll-free number with predefined given information through spoken language and, as such, could be a way to communicate with Maasai in their language, who often have low literacy and do not quite understand the Swahili language [15].

Based on this, we explored the effect and feasibility of an IVRC system for family planning education among Maasai couples of reproductive age living in the Esilalei ward, Monduli District, in the Arusha Region. The main objective of this study was to explore the change in knowledge about, and use of, family planning methods after implementing an educational interactive voice response calling system about family planning and contraceptives for Maasai. Specific objectives were (1) to investigate whether the IVRC system improved knowledge about contraceptives, (2) to examine whether the system increased access to family planning care, (3) to determine whether the rate of contraceptive use improved, and (4) to establish whether the system was acceptable and feasible in use.

## Methodology

### Study design

We conducted participatory action research using mixed methods, including quantitative and qualitative approaches for data collection. This serial design consisted of a baseline assessment, development of the digital system, deployment of the system, and final assessment among Maasai using several data sources described below. The Kilimanjaro Christian Medical College Research Ethics and Review Committee (CRERC) and the National Health Research Ethics Sub-Committee (NatHREC) of Tanzania approved the study. The study was conducted in accordance with the Declaration of Helsinki.

### Study area and population

We conducted our study among married Maasai couples (i.e. a husband with one of his wives) in the Esilalei ward in Monduli District, Arusha Region, Tanzania. We selected Maasai men using convenience sampling. The husband chose one of his wives to participate in the study. The ward consists of three villages: Esilalei, Losirwa, and Oltukai, and most of the population consists of the Maasai tribe. One dispensary in the ward provides family planning services, including counselling and the provision of some contraceptive methods, including oral contraceptive pills, injections, and implants. Additionally, we enrolled healthcare workers for an in-depth interview.

### Study procedures

**Baseline assessment.** We did a baseline assessment in January and February of the year 2019. Maasai couples were enrolled by firstly screening their eligibility, i.e. if they were a couple from the Esilalei ward (husband and one of his wives), aged between 12 and 65 (age of wife <50), willing to receive IVRC and able to understand and willing to sign the informed consent document. If eligible, we extensively explained the study to them and asked for written informed consent to participate. We used the local Maasai language, Maa, to communicate with participants through our Maasai interpreters. If written informed consent was not possible, we collected thumbprints after approval to participate. The husband and wife gave consent to be part of the study if they were above 18 years. For men under 18 years, we asked consent from their father/elder, the legally acceptable representative (LAR), in addition to assent from

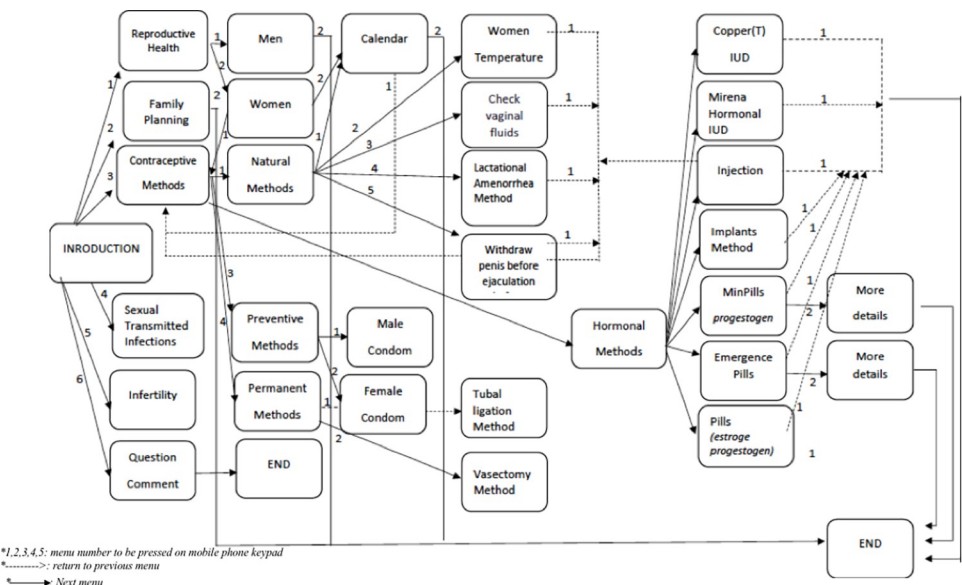

**Fig 1. Provides the details of IVRC content flow.**

themselves. For women under 18 years, we asked both the husband and the elder to consent, and we asked for assent from the women. Following written informed and oral consent, participants were interviewed using a structured questionnaire to assess knowledge about and use of contraceptives. Furthermore, we abstracted information on family planning clinic visits from available medical records at Esilalei dispensary and Mto-wa-Mbu health center to determine the number of clinic visits and prescribed contraceptive methods in the year before the implementation of Embiotishu (January 2018-February 2019).

**Development and implementation of Embiotishu.** We developed an interactive voice response calling (IVRC) system called Embiotishu.Fig 1 provides the details of IVRC content flow.The system was accessible for Maasai by calling a toll-free number. The Embiotishu (health in Maa) system was developed based on the information obtained during the basic assessment with the guidance of Tanzanian guidelines for family planning. Furthermore, several meetings were conducted with Maasai couples, local health care providers, and study consultant obstetrician gynecologists to set the criteria and identify the knowledge gap in reproductive health and the contents of the system. Texts were translated to Maa and categorized based on themes, including reproductive health, family planning, and STIs/HIV/AIDS. We created audio messages by recording the spoken texts in Maa language and loaded them into the Embiotishu system. A menu was developed for the contents and loaded in Embiotishu. Each menu item was assigned a number (e.g. 1,2,3,4) that study participants (Maasai couples) could press on their mobile phone keypad when selecting a particular item. When a study participant calls Embiotishu (s)he hears: "Welcome to Embiotishu, a programme to inform you about reproductive health and family planning. For reproductive health, press 1; for information about contraceptives, press 2; for information about family planning, press 3'. Once the IVRC system recognized the number entered, it automatically routed the request to the particular pre-recorded content. The IVRC system allows multiple requests from different users in the same call. Before the deployment of Embiotishu, we conducted seminars with both parties (Maasai and Health providers) to introduce the system. During the seminars, we did role plays to instruct users about the system, and we provided the toll-free number. Participants without a cellphone were provided with one. We deployed the system for 20 months.

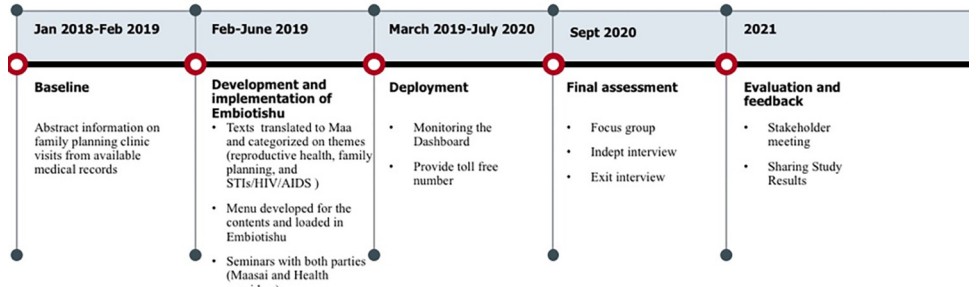

**Fig 2. describes the study course and duration of the study phases.**

**Final assessment.**   After twenty months of follow-up, we repeated the survey *(S1 Appendix)* to investigate the study participants' knowledge of family planning methods (see Fig 2). We also continued the medical abstraction to explore the trend of family planning utilization in the clinic (access to care) since deploying Embiotishu (March 2019-July 2020). We conducted focus group discussions using semi-structured topic guides with Maasai couples to investigate the acceptability and feasibility of Embiotishu and to explore the knowledge and use of FP. To investigate the number of times the system has been used and which topic was mainly accessed, we explored the number of incoming calls to Embiotishu and the number of chosen menu items. The illustration of the study procedures has been described below.

## Data collection tools

**Survey with Maasai.**   We administered a semi-structured questionnaire in face-to-face interviews during the baseline and final assessment among Maasai couples. Swahili or Maa language was used based on the preference of the interviewee. The questionnaire contained questions on socio-demographic characteristics, marital status, family composition, reproductive history, knowledge about specific contraceptive methods and use of contraceptives. In the final assessment, we added questions to explore their experience with the Embiotishu system. Data were collected on a paper form and entered in the REDcap database, and stored on the local server [16].

**Medical records abstraction.**   A medical records abstraction form was developed based on the standard clinic card for family planning (Reproductive Child and Health card no.5). We collected data from medical files of the family planning clinics of Esilalei Dispensary and Mto-wa-Mbu Health Centre. The following information was collected; clinic visit date, reproductive history, current contraceptive use, and prescribed contraceptives. Data was collected in paper form, entered by trained staff in the REDCAP database directly, and stored on a local server.

**FGD with Maasai couples.**   We conducted focus group discussions in the final assessment using a topic guide by a trained qualitative researcher using the Maasai project leaders as interpreters. The topics discussed were: (1) knowledge about the Embiotishu system, (2) "How Embiotishu changed Maasai life", (3) knowledge about and use of family planning methods, (4) knowledge about the reproductive system, (5) knowledge about sexually transmitted diseases and (6) suggested improvements for the Embiotishu system. Discussions were done in the Maasai language with continuous translation and interaction between the researcher and interpreters in separate groups of men and women.

**IDI with health care providers.**   We conducted in-Depth interviews with Health Care providers involved in the family planning services during the baseline to understand and

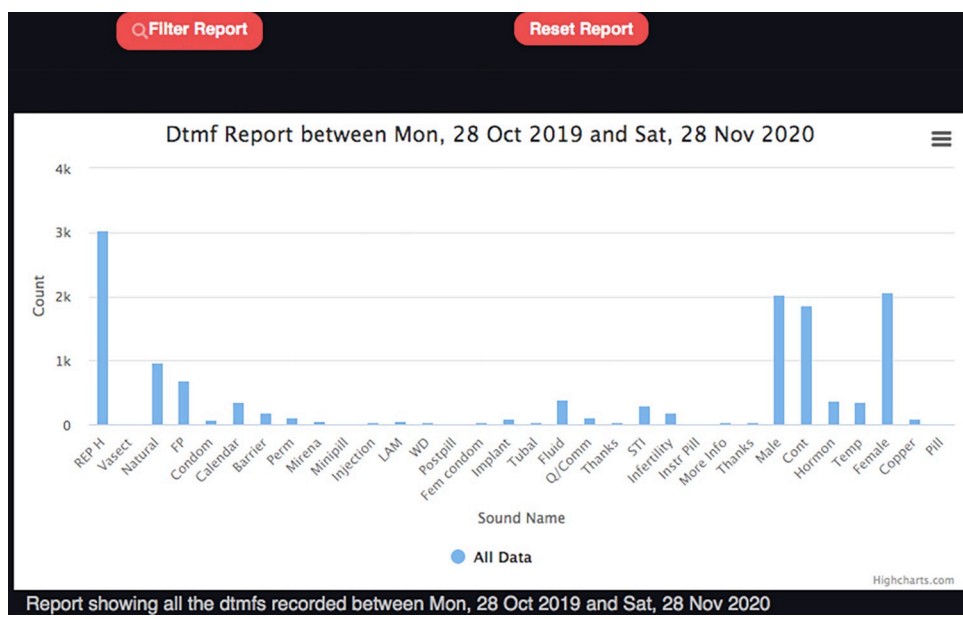

**Fig 3. The distribution of selected topics on the system dashboard.**

obtain their views on how the Maasai know, access, and use contraceptives to contribute to the development of the Embiotishu system. Later IDIs were conducted during the final assessment among HCWs who participated in the study to explore their view on the knowledge of family planning methods among Maasai people after Embiotishu. We used semi-structured interview guides to assess (1) knowledge about contraceptives among Maasai, (2) access to family planning methods, (3) improved use of contraceptives because of Embiotishu and (4) the acceptability of the Embiotishu system.

**Embiotishu system data.** Incoming calls and responses (menu items) data of Embiotishu were obtained automatically via an online dashboard as illustrates in Fig 3. To access the information, we provided login credentials such as passwords and usernames to authorized users. The information that was recorded included: caller ID, information type, time, date, and call-frequencies. We extracted this information from the system in the form of graphs.

To gain feedback from Maasai couples about Embiotishu, we conducted interviews with semi-structured questionnaires during the final assessment among couples. The questions included their broad experience with the system, preferred topic, challenges, and recommendations to improve the Embiotishu system.

## Data analysis

For quantitative data, we conducted descriptive analyses with SPSS v.27 to investigate changes and trends in knowledge and use. Data from the Embiotishu was displayed in graphs using an excel spreadsheet program to create frequency tables.

We transcribed and translated qualitative data into English. Transcripts were read and reread by PM, TM, and MS and imported into MaxQDA 2020. We created memos based on the transcripts from where we identified codes for organizing our qualitative data. We conducted a thematic framework analysis (PM) [17] with an inductive approach and used the coded transcripts to obtain different themes and subthemes. The themes were related to knowledge, access, and use of family planning methods and the acceptability and feasibility of the Embiotishu system.

To answer objective one on knowledge, we investigated with data from the survey whether the percentage of participants knowing about different contraceptive methods ('yes'-answers) changed before and after using Embiotishu by conducting McNemar tests. In addition, to get an overall score on the knowledge, we summed knowledge about all contraceptives ('yes'-answers being 1 point) into one score. We conducted T-tests to investigate differences in scores before and after the system's deployment for both men and women. Furthermore, we analyzed data from the final assessment FGD and IDI by identifying themes related to knowledge about family planning methods.

To answer the second objective on access to family planning methods, we examined whether the number of clinic visits for family planning between baseline and final assessment increased. We used descriptive analysis to describe the percentage of the data from the survey.

To answer the third objective on the use of family planning methods, we investigated the difference in the use of contraceptives between baseline and final assessment with a Chi-square test. In addition, we identified themes related to the use of family planning methods discussed in the FGD.

To answer the fourth objective on the acceptability and feasibility of Embiotishu, we identified themes related to these topics from both the FGD and IDI. In addition, we descriptively analyzed data about the experience with the system from selected topics recorded by the system and the survey in the final assessment.

## Results

### Study population

We recruited 76 Maasai couples and we interviewed them during the baseline assessment. The mean age was 25[SD:5.9] years for women and 32.5[SD:10.5] for men. Of women, 65% had no formal education, 27% completed primary education, while 35% of men had no formal education, and 47% completed primary education. Also, 63% and 68% of women and men, were Christian. No one was employed among women, while 9% of men were employed. Other men were engaged in farming activities for income. (*Table 1*)

We held four FGDs, with 10 to 13 Maasai in each group. Two groups were Maasai women, and the other two were Maasai men. Several themes were deducted from the discussion, including (1) knowledge about contraceptives among Maasai, (2) access to family planning methods, (3) improved use of contraceptives because of Embiotishu and (4) the acceptability of the Embiotishu system. Descriptions of the themes and examples are described in the (*S1 Appendix*)

### Knowledge about family planning and contraceptives

Table 2 describes the difference in knowledge about contraceptives before and after the implementation of Embiotishu among women and men. For women, knowledge about all contraceptives significantly increased (p<0.05). The highest increase was for knowledge about the intra-uterine device (IUD), condoms, and Lactational Amenorrhea Method. For men, it only increased for vasectomy, injection, implants, contraceptive pills, condoms, and withdrawal (p<0.05)

The overall knowledge of contraceptives between men and women before and after the implementation of Embiotishu increased significantly (P-value<0.001). (*Table 3*)

The findings from qualitative data indicated that the Embiotishu system increases the knowledge about FP among Maasai. As a result, it positively influences their attitude towards practicing family planning. This was witnessed when they stated:

**Table 1. Socio-demographic characteristics of Maasai.**

| Characteristics | Women | | Men | |
|---|---|---|---|---|
| | Number | Percent, % | Number | Percent, % |
| Total | 76 | 100% | 76 | 100% |
| Age (years) | | | | |
| Mean [SD] | 25 [5.9] | | 33 [10.1] | |
| Level of education | | | | |
| No formal education | 50 | 65% | 27 | 36% |
| Some primary education | 4 | 4% | 8 | 10% |
| Primary education | 21 | 27% | 34 | 45% |
| Secondary education | 1 | 1% | 6 | 8% |
| Higher education | 1 | 1% | 0 | 0% |
| Missing | 1 | 1% | 1 | 1% |
| Literacy rate (can read Swahili) | 27 | 36% | 48 | 63% |
| Does own a cell phone | 31 | 41% | 60 | 79% |
| Employed | 0 | 0% | 7 | 9% |
| Husband has more than one wife | 31 | 41% | 33 | 43% |
| Wife Included in the study (1st 2nd 3rd) | | | | |
| 1st wife | 15 | 20% | - | - |
| 2nd wife | 19 | 25% | - | - |
| 3rd wife and more | 10 | 12% | - | - |
| Age started to live with a partner | | | 23 [4.8] | |
| Mean [SD] | 17 [5.9] | | | |
| Age when 1st child was born | | | 27 [5.8] | |
| Mean [SD] | 18 [4.1] | | | |

*"We have seen the use of Embiotishu numbers is clearer and easier to understand as it teaches us about family planning, so now we know family planning and many other things" (Male, 28 years)*

**Table 2. Knowledge about contraceptives.**

| Knowing about contraceptive | Women | | | Men | | |
|---|---|---|---|---|---|---|
| | Basic assessment n (%) | Final assessment n (%) | P-value | Basic Assessment n (%) | Final Assessment n (%) | P-value |
| Tubal ligation | 24(31.6) | 55(72.4) | <0.001 | 41(53.9) | 42(55.3) | 0.6 |
| Vasectomy | 10(13.2) | 47(61.8) | <0.001 | 12(15.8) | 23(30.3) | 0.03 |
| Loop/intra-uterine device (IUD)-Mirena/Copper-T | 27(35.5) | 64(84.2) | <0.001 | 58(76.3) | 63(82.9) | 0.013 |
| Injection (Depo-provera) | 45(59.2) | 69(90.8) | <0.001 | 57(75.0) | 66(86.8) | <0.001 |
| Implants (Implanon) | 50(65.8) | 69(90.8) | <0.001 | 60(78.9) | 66(86.8) | <0.001 |
| Oral contraceptive pills | 44(57.9) | 65(85.5) | <0.001 | 50(65.8) | 62(81.6) | <0.001 |
| Condoms | 23(33.3) | 59(77.6) | <0.001 | 54(71.1) | 65(85.5) | <0.001 |
| Emergency contraceptives (morning-after-pill) | 25(32.9) | 59(77.6) | <0.001 | 38(50.0) | 41(53.9) | 0.3 |
| Cycle beads to count the days | 8(10.5) | 57(75) | <0.001 | 21(27.6) | 27(35.5) | 0.2 |
| Lactational Amenorrhea Method (LAM) | 30(39.5) | 62(81.6) | <0.001 | 35(46.1) | 24(31.6) | 0.18 |
| Calendar rhythm method | 51(67.1) | 62(81.6) | 0.003 | 71(93.4) | 66(86.8) | 0.1 |
| Withdrawal method | 52(68.4) | 64(84.2) | <0.001 | 56(73.7) | 67(88.2) | 0.001 |

**Table 3.** Difference in knowledge about all contraceptive methods (sum score) before and after Embiotishu.

| Gender | Basic assessment mean (SD)–Sum of all contraceptive methods | Final assessment mean (SD)–Sum of all contraceptive methods | P-value |
|--------|------------------------------------------------------------|------------------------------------------------------------|---------|
| Women | 5.0(3.2) | 10.6(2.3) | <0.001 |
| Men | 7.5(2.8) | 9.0(1.8) | <0.001 |

The information about family planning methods in the Embiotishu system enabled Maasai couples to make a conscious effort to limit or space the number of children they have and reduce fertility. It was evidenced by the following:

*"We have heard a lot of good things in the Embiotishu system, which in general are focusing on putting us in the position of being able to plan several children in our families." (Female, 23 years)*

Family planning gives knowledge of the reproductive system. It advances the health of women and children by lowering the proportion of pregnancies that could be high risk and reducing the number of unplanned pregnancies and births. One was aware of family planning benefits and was quoted as:

*"And the other thing you get from it is to know the parts of Male and Female reproductive systems, to understand that you can stay for how long or some days without having sex with your wife. Such things are described, which gives us more education." (Male, 35 years)*

Another respondent said:

*"And the things that are being discussed, for real, if someone decides to consider it, will help you because it tells you many things. Particularly about the female and male reproductive system and understanding your wife's menstrual cycle." (Male, 40 years)*

From the FGDs, we found that the knowledge gained from the system among Maasai people accomplishes what they require and allows them to plan their families well. In previous days, they were staying far from their wives, worrying about having a larger number of children, which they had not planned yet. This was reported as follows:

*"The calendar method is good because it has no hormones, and it will show you that she has cheated because she will get pregnant." (Male, 55years)*

### Access to family planning care

In the year before the implementation of Embiotishu (2018), the number of visits for family planning methods was lower (N = 22) compared to the years after implementation in 2019 (N = 84) and 2020 (N = 214) as illustrates in Fig 4.

Findings from qualitative data show that the Embiotishu system increased the perception concerning family planning. It increases the number of clinic visits. Also, family planning provides benefits to families and their personal development. One of the nurses we interviewed told us:

*"Yes, the system made changes in our work because customers seem to understand well the Family Planning system through the details they already got." (Nurse, 44 years)*

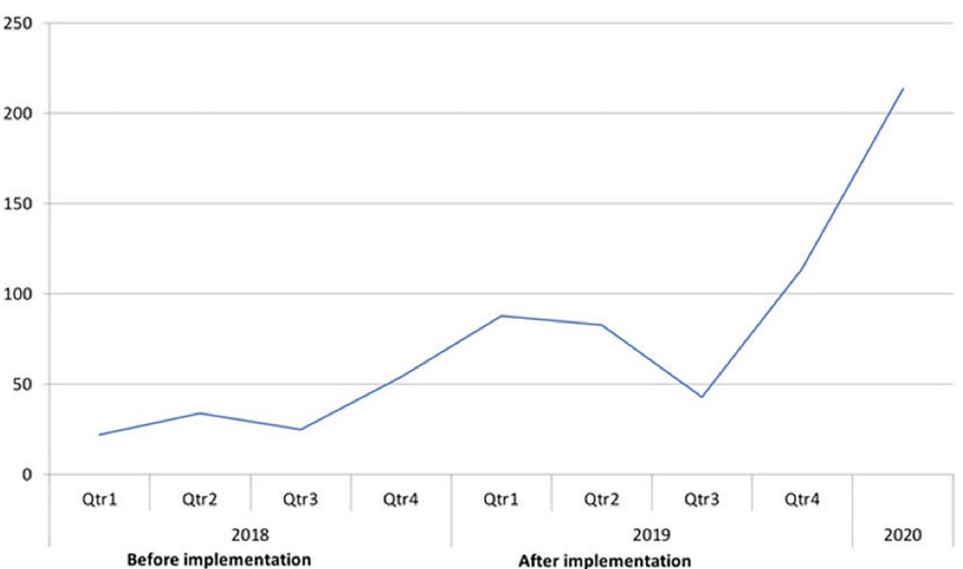

**Fig 4. Number of family planning clinic visits per quarter per year.**

Another nurse added and said:

*"These days, men around here realize the importance of family planning. Some even accompany their wives to the clinic." (Nurse, 43 years)*

Education introduced among Maasai people concerning family planning increased awareness of the use of contraceptives and increased availability. It is an essential factor in fertility decline. One of the participants said:

*"So, if people get awareness about using contraceptives like paracetamol, if you feel pain, you can send a child to buy it at the shop. It will be easy to select several points to put boxes of condoms, and anybody who wants it may go there and pick it." (Male, 55years)*

## Use of family planning methods

Out of the 76 couples interviewed before the implementation, only 26(27%) couples revealed they used contraceptive methods before Embiotishu. After implementation, 69(73%) couples revealed they used contraceptives (p<0.001).

Before and after the implementation of Embiotishu, from the medical records (see Fig 5), we saw that implant was the most prescribed family planning method from 24(3%) before to 447 (65%), followed by injection from 9(1%) to 105 (15%), condoms from zero to 47(7%) and contraceptive pills from zero to 36 (5%).

Qualitative data showed that informing Maasai families about FP was associated with acceptance of family planning in terms of reducing family size and the ability to use contraceptives. Most of the Maasai indicated their perception has increased that after being educated, and they can handle their families well. This was supported:

*"Yes, it is true people have changed. For example, to my side up to this stage am now, we think that if we have five children and they need to go to school, it makes us think about spacing to take care of those we already have. So, it is nice, and it makes people change their perception." (Male, 36 years)*

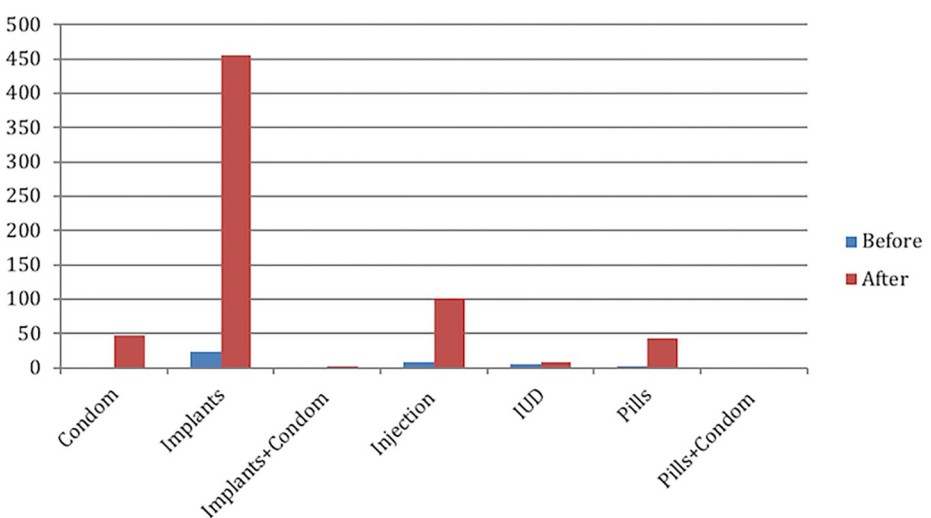

**Fig 5. The distribution of selected topics.**

The acceptability of contraceptives was also accompanied by the possible choice of natural contraceptives, which also proved beneficial. The woman commented and said:

*"We are no longer giving birth too much because we know family planning. Because through the Embiotishu number, we were told things to do, and these things are practical." (Female, 43 years)*

This is also supported by one of the group members saying that:

*"Contraceptives methods make us not to have unexpected pregnancies." (Female, 18 years)*

This was also explained when we were interviewing nurses. It was reported that:

*"Yes, they tell each other this is good, you can stay with it up to five years, so it is good depending on the number of children they have." (Nurse, 29 years)*

## Acceptability and feasibility of Embiotishu

**Feedback from Maasai.**   Out of the participants (N = 152) enrolled in the study, we interviewed 105 about their experience with the system; 57(95%) of men and 48(91%) of women. Almost all men, 49(82%) and 37(74%) women, indicated the system was well understood and easy to navigate through the information. Ninety-four percent of women (N = 53) and eighty-seven percent of men(N = 42) indicated that the system performed well. The majority, 94 (62%), indicated that the most interesting topics were reproductive health and contraceptive methods. Furthermore, others recommended topics related to sexually transmitted diseases and infertility to be added to Embiotishu. A few participants, 18(12%), indicated communicating with healthcare workers about information from Embiotishu.

During FGD discussions, Maasai described that the dry season causes substantial migration of the family and their animal mainly to search for water and food.

As mentioned by one participant:

*"When a dry season comes like now, we are only concerned about where to get green grass and water for our animals and family. We cannot think about family planning issues or access information in Embiotishu because we are struggling here and there in search of water." (Male, Age 45 years)*

Embiotishu proved to be of advantage also to the marital relationship. This was when the marital sexual relationship was no longer limited during the lactation period. One participant mentioned:

*"Embiotishu has brought us many very good things, and this is because now a person can dial that number and listen. You know, before, we did not have sex with our wives while they were still lactating. If a woman has lactated for five years, you also stay for those years without having sex with her, but right now, things have changed; when we go to the market, you come with condoms, and then you can have sex with her even if she has three or five months after giving birth." (Male,25 years)*

Another respondent shared;

*"I always call the Embiotishu number seven times a day." (Female, 45years)*

The same respondent reported that:

*"I always like to dial the Embiotishu number and press one. It tells you about your reproductive system in general, so you will know that this is good because it instructs you in a direction that when you follow it, you will not fall or it will not let you down." (Female, 45 years)*

**Access to contraceptive contents in the Embiotishu system.** We recorded a total number of 24,033 incoming calls, including missed calls, calls in which no topic was selected, and questions in the system in 20 months from 2019 to 2021 as seen in Fig 6. Out of those, 14,547 topics were selected. The highest number of calls was recorded in March, April, and May 2020.

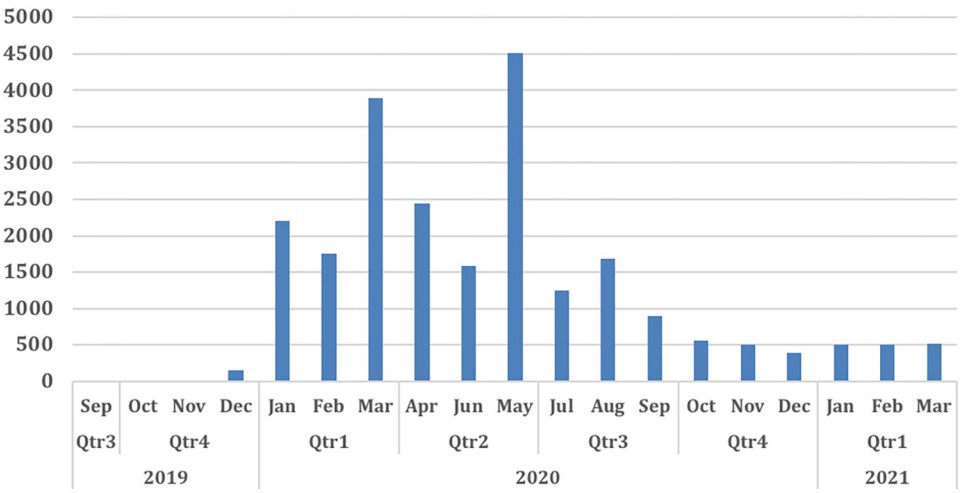

**Fig 6. The trend of calls in Embiotishu.**

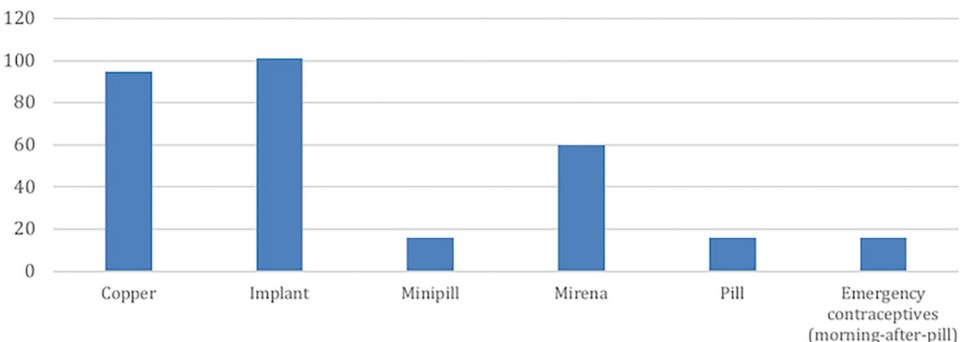

**Fig 7. Access to hormonal contraceptive content in IVRC.**

**Modern contraceptives.** In 400 calls, the topic of hormonal contraceptives was selected as shown Fig 7. Among those calls, in 25% implant was selected, 24% copper IUD, 18% Mirena (hormonal IUD), 4% pills (estrogen and progestogen) and 4% minipill (progestogen hormone only) were selected.

Out of those 197 calls about non-hormonal contraceptives, the barrier methods topic male condoms were selected 65 times (33%) and female condoms 40 times (20%) as shown in Fig 8. For permanent methods, tubal ligation was selected 42 times (21%), followed by vasectomy (12%).

**Natural contraceptives.** In a total of 1,376 calls, the topic of natural contraceptives was selected (see Fig 9). The most selected natural contraceptive was 'observing of vaginal fluid' with 441 times (33%), followed by monitoring the women's temperature with 440 times (32%) and avoiding sex on danger days 391 times (28%). In a few calls, withdrawing the penis before ejaculation and LAM (exclusively breastfeeding) methods were selected.

## Discussion

Our study shows that the IVRC system is highly feasible and acceptable for improving knowledge about and using contraceptives, family planning, and sexual and reproductive health. We showed this through the survey in which knowledge about the number of family planning

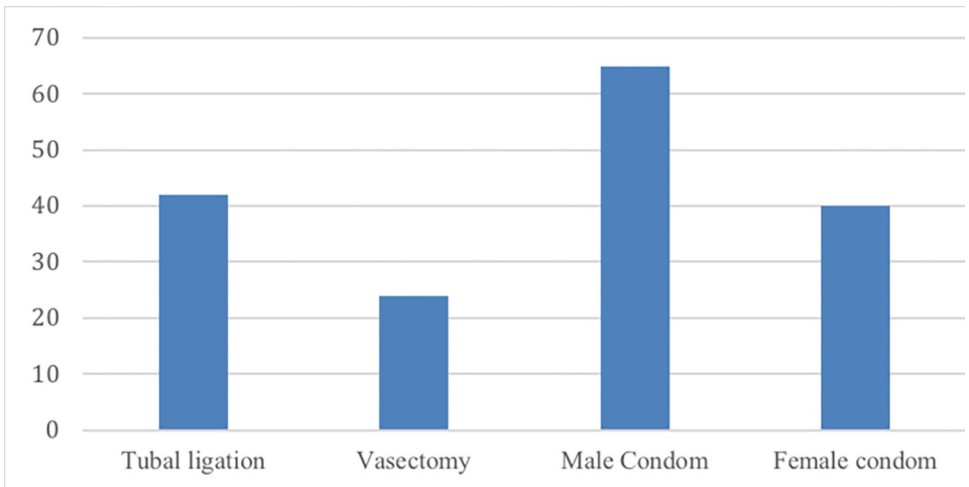

**Fig 8. Non-hormonal contraceptives.**

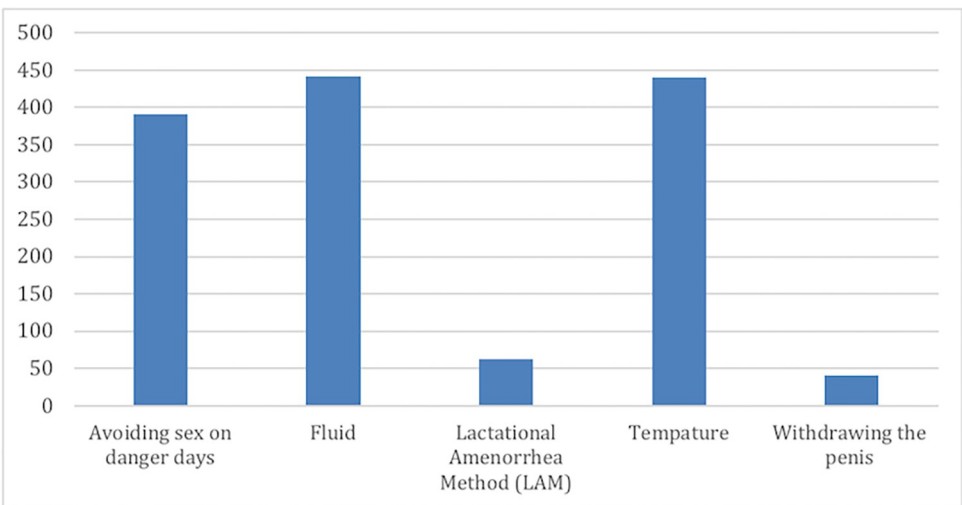

**Fig 9. Natural contraceptive use.**

methods increased significantly, and the use of contraceptive methods increased after our program. We observed similar results during focus group discussions. Data from the discussions showed that implementing the Embiotishu system positively changed the Maasai attitudes towards practicing family planning. Furthermore, participants revealed that the education provided by IVRC makes them able to identify effective contraceptives. Medical abstraction results showed that the visits for family planning increased after the implementing Embiotishu. Nurses revealed that Embiotishu changed the Maasai perception concerning family planning, which resulted in men's and women's' understanding of the importance of family planning. Therefore, the system has the potential to limit the number of children and spacing between children and reduce fertility.

These results are comparable to a study conducted in Uganda which showed that learning about family planning among women living along lake Victoria increased their ability to plan for their families [18]. Another study in Kenya describes that knowledge of contraceptives among pastoralists improves access to family planning and decreases the burden on the household due to the higher cost of living [19]. This was related to the study by Allegretti et al., which showed that pastoral household heads find it challenging to support larger families due to the rising demand for money, which does not match the available cash income sources [20]. Further, this was also stated by Leah et al., who found that access to reproductive health knowledge prevents unwanted pregnancies, especially among adolescents, and prevents sexually transmitted diseases and AIDS [21].

A prior study has shown that Maasai men have a negative perception of family planning because it will lower the birth rate and cause infertility. Also, Maasai believed that having many children was a sign of wealth and prestige in the community [22]. However, our study shows that Maasai were interested to use family planning and apply child spacing in order to satisfy the family's needs including education, shelter, and food.

During the discussion, Maasai women expressed that they were empowered by Embiotishu, particularly on modern contraceptive topics regarding family planning, which led them to have a firm decision on child spacing. In addition, IVRC positively influenced their attitude towards practicing family planning because the content was more explicit and easier to understand.

These factors are also substantiated by the findings from medical records findings showing that modern implants were the most prescribed family planning method.

A previous study conducted in Tanzania described modern contraceptive use as contributing to the low rate of unintended pregnancy among Married Maasai women who felt socially pressured to bear many children as desired by their husbands or family [23]. A study from Kenya has shown that education about contraceptives empowered women and made them more aware of better decisions on contraceptive use [24]. Furthermore, few Maasai selected information about condoms and hormonal contraceptives in Embiotishu. As described in the previous studies, the sociocultural context contributes to a negative perception as Maasai men believe that using condoms leads to a waste of semen. Also, the use of hormonal contraceptives was believed to cause cancer, increase weight and kill women's eggs (i.e. cause infertility) [5,24]. Therefore, opinions of different age groups and professionals will contribute to the implementation.

Another finding was the high volume of incoming calls to the system in certain months. It was revealed during the focus group discussions to be strongly related to the rainy season. Many areas become green for grazing the cattle, and as a result, Maasai men spend more time with their families and wives at bomas. A study in Kenya described a similar trend during the rainy season, in which Maasai men spend time on other issues such as traditional ceremonies, selling or buying cattle, and expanding their families [22].

## Limitations

We conducted the study in only one village ward, and the number of participants was small. This makes the representativeness of the total Maasai population in Tanzania questionable. As Esilalei is situated in a relatively accessible area with some basic health care services, we expect that the population is not representative of the total Maasai population who live in the less accessible areas. In addition, we have not conducted an intervention study comparing results between people exposed and not exposed to Embiotishu. We can, therefore, not conclude that our intervention is effective. However, our study provides enough proof for the intervention to be investigated in a larger trial to determine its effectiveness.

## Conclusion and future work

The project's development was based on a participatory methodology whereby the Maasai community was involved in the early stages of the research by sharing their knowledge, ideas, values, and opinions. This helped us to familiarize ourselves with indigenous knowledge, challenges, and barriers which guided us to tailor the IVRC content to specific targets. In addition, using a mixed-methods approach and different data sources, we obtained rich data, which allowed us to answer the objectives effectively and gave good insight into the context of our data.

Our study has shown that the IVRC system led to an improvement in knowledge about the use of contraceptives. Embiotishu also appeared to be feasible and acceptable among the Maasai. Furthermore, it has the potential to improve access to health information. However, it is essential to consider the norms and culture of the Maasai community on the topics of family planning, reproductive health, and contraceptives. Therefore, it is relevant to incorporate all attributes related to the socio-cultural aspects of Maasai to ensure that future interventions effectively empower the Maasai community. Furthermore, the rapidly growing migration of Maasai to urban areas in recent years and mobile technology services facilitate sharing of information in a distant location. Future studies should continue to explore the effect of this digital technology for continuous advocacy of the use of contraceptives and family planning among the Maasai community in randomized controlled settings.

## Supporting information

**S1 Appendix. Survey on Investigating the knowledge about family planning.**
(PDF)

## Acknowledgments

We would firstly like to thank Voice.global for financially supporting our study under the linking and learning grant for Tanzania. We thank our research assistants from the Maasai community, Mbayani Lemkoko, Esta Mbayani, Elizabeth Morine, and Rueben Moitiko who assisted us with collecting all data during interviews and focus group discussions. We also thank our partner in this project, the African Roots Foundation, through Chris Pilley. Further, we also thank our colleagues who were involved in the proposal development and preliminary focus group discussions which were Godfrey Kisigo, Iraseni Swai, and Martha Oshosen. Lastly, we thank the participants for providing valuable and sensitive information.

## Author Contributions

**Conceptualization:** Kennedy Ngowi, Benson Mtesha, Krisanta Kiwango, Ester Kiwelu, Benjamin C. Shayo, Eusebious Maro, I. Marion Sumari-de Boer.

**Data curation:** Kennedy Ngowi, Benson Mtesha, Jacqueline Kwayu, Tauta Mappi, Ester Kiwelu, Benjamin C. Shayo, Eusebious Maro, I. Marion Sumari-de Boer.

**Formal analysis:** Kennedy Ngowi, Perry Msoka, Benson Mtesha, Tauta Mappi, I. Marion Sumari-de Boer.

**Funding acquisition:** I. Marion Sumari-de Boer.

**Investigation:** I. Marion Sumari-de Boer.

**Methodology:** Kennedy Ngowi, Benson Mtesha, Jacqueline Kwayu, Tauta Mappi, Benjamin C. Shayo, Eusebious Maro, I. Marion Sumari-de Boer.

**Project administration:** Kennedy Ngowi, I. Marion Sumari-de Boer.

**Resources:** Titus Mmasi, Aifello Sichalwe.

**Software:** Kennedy Ngowi.

**Supervision:** I. Marion Sumari-de Boer.

**Validation:** I. Marion Sumari-de Boer.

**Visualization:** Perry Msoka, Benjamin C. Shayo, Eusebious Maro.

**Writing – original draft:** Kennedy Ngowi, Perry Msoka.

**Writing – review & editing:** Kennedy Ngowi, Benson Mtesha, Jacqueline Kwayu, Tauta Mappi, Krisanta Kiwango, Benjamin C. Shayo, I. Marion Sumari-de Boer.

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
