## [Decision Letter · Decision Letter 0]

3 Aug 2022

PDIG-D-22-00061

"The phone number is telling us good things which we didn’t know before.” Use of Interactive Voice Response Calling for Improving knowledge and uptake of family planning methods among Maasai in Tanzania

PLOS Digital Health

Dear Dr. Ngowi,

Thank you for submitting your manuscript to PLOS Digital Health. After careful consideration, we feel that it has merit but does not fully meet PLOS Digital Health's publication criteria as it currently stands. Therefore, we invite you to submit a revised version of the manuscript that addresses the points raised during the review process.

Please submit your revised manuscript within 60 days Oct 02 2022 11:59PM. If you will need more time than this to complete your revisions, please reply to this message or contact the journal office at digitalhealth@plos.org. Please include the following items when submitting your revised manuscript:

We look forward to receiving your revised manuscript.

Kind regards,

Yuan Lai, Ph.D.

Academic Editor

PLOS Digital Health

Journal Requirements:

1. Please amend your detailed online Financial Disclosure statement. This is published with the article. It must therefore be completed in full sentences and contain the exact wording you wish to be published.

2. Please update your online Competing Interests statement. If you have no competing interests to declare, please state: “The authors have declared that no competing interests exist.”

3. In the online submission form, you indicated that your data will be submitted to a repository upon acceptance. We strongly recommend all authors deposit their data before acceptance, as the process can be lengthy and hold up publication timelines. Please note that, though access restrictions are acceptable now, your entire data will need to be made freely accessible if your manuscript is accepted for publication. This policy applies to all data except where public deposition would breach compliance with the protocol approved by your research ethics board. If you are unable to adhere to our open data policy, please kindly revise your statement to explain your reasoning and we will seek the editor's input on an exemption. Please be assured that, once you have provided your new statement, the assessment of your exemption will not hold up the peer review process.

4. Please provide separate figure files in .tif or .eps format and remove any figures embedded in your manuscript file. Please also ensure that all files are under our size limit of 10MB.

For more information about how to convert your figure files please see our guidelines: https://journals.plos.org/digitalhealth/s/figures

Additional Editor Comments (if provided):

Reviewers' comments:

Reviewer's Responses to Questions

**Comments to the Author**

1. Does this manuscript meet PLOS Digital Health’s publication criteria? Is the manuscript technically sound, and do the data support the conclusions? The manuscript must describe methodologically and ethically rigorous research with conclusions that are appropriately drawn based on the data presented.

Reviewer #1: Yes

Reviewer #2: Yes

Reviewer #3: Partly

2. Has the statistical analysis been performed appropriately and rigorously?

Reviewer #1: Yes

Reviewer #2: Yes

Reviewer #3: Yes

3. Have the authors made all data underlying the findings in their manuscript fully available (please refer to the Data Availability Statement at the start of the manuscript PDF file)?

Reviewer #1: Yes

Reviewer #2: Yes

Reviewer #3: No

4. Is the manuscript presented in an intelligible fashion and written in standard English?

Reviewer #1: No

Reviewer #2: Yes

Reviewer #3: No

5. Review Comments to the Author

Reviewer #1: Dear Authors, this is great work in addressing those individuals as it helps to do further research for future. therefore try to amend the comments given in the source document then proceed accordingly.

Reviewer #2: There is no literature review, at least to show the gap between the previous studies & to show what is the the new in the current study

References are very few

Limitation & strength should be moved to conclusion and the title should be conclusion & Future work

Methodology should be reformulated, to many speech which need to be minimize 

It seems that it’s a copy from PhD or MSc thesis , so it’s need to be formulated

Reviewer #3: This research article shows how digital health solutions can be applied to increase knowledge and improve health care in rural and undersupplied areas. Furthermore, the authors found a smart solution in the Interactive Voise Response (IVR) system to overcome literacy barriers. The different methodologies and outcome sources used in implementing and evaluating the IVR system raise the quality of this research. 

Nevertheless, there are some concerns regarding formatting and language, completeness of the methods, data availability, figures, precision regarding the results and missing discussion points. These are described in the following.

Formatting and language: The formatting of the text is unclean in many places. There are double spaces, misplaced commas and dots, unfinished phrases (e.g. Abstract, last phrase of the results), and citation outside the phrase in the manuscript. Moreover, the manuscript should be revised concerning language and grammar before publication. 

Methods:

- Please explain the study design (participatory action research using mixed methods) more deeply for non-experts in this field.

- A statement regarding the Declaration of Helsinki is missing. 

- How were health care workers selected for the study? How many were included? Are nurses and health care workers the same participant group? 

- In the study procedures it is stated that Maasai couples were screened for their eligibility. What are the eligibility criteria? 

- Add a protocol of the semi-structured questionnaire to the supporting information or describe the relevant questions that were used for data analysis.

- Was the medical records abstraction collected for the population in general or specifically for the participants? 

- Explain or cite thematic framework analysis.

- Study course: The study course and duration of study phases (deployment and development of the IVR, assessment time points) is not clearly defined. A figure could be helpful here. 

Data availability: Data is not available yet.

Figures:

- Figure 1: Please explain the figure in the legend and in the text. What do arrows mean? Difference between dashed and solid line? What do numbers mean? How does the IVR work? 

- All figures and tables: Please explain figures and tables in the respective legend. 

Results:

- It is not clear which data from which source was used for which analysis. E.g., family planning methods: How did couples revealed to use contraceptives? Which question was asked?

- Table 1: What do the numbers in the religion cell mean? Was there no information available for the remaining percentage?

- Table 1: Not all participants owned a cell phone. How did those without a cell phone take part in the study?

- Table 2: How was the knowledge assessed exactly? Which questions were asked? 

- Table 2: It looks like there is a difference between women and men, regarding the knowledge at baseline and the increase in knowledge from baseline to final assessment. Could be an interesting topic for the discussion. Did women used the IVR system more and accepted it better?

- Table 2: For men, there are some cells where the knowledge decreased. How is that possible? Do you have an explanation for that?

- Table 3: Please explain how the values are composed. 

- Qualitative analysis: The presentation of the results of the qualitative analysis are not sufficient. Which themes and subthemes (according to Data analysis section) were identified? How many statements were allocated to which themes? Please give the reader more information here. 

- Access to family planning methods: The number of clinical visits is only descriptively increasing. Can this also be statistically proven?

- Page 18 first statement: From a woman or a men? Contradictory information here. 

- Acceptability and feasibility: Please indicate exact numbers or percentages here. Almost or nearly all is not precise enough. 

- Access to contraceptive contents: What is a missed call? A call where no topic was selected?

- A figure where the distribution of selected topics or categories among the calls is depicted could be eventually helpful for the reader. 

- Page 21: “Out of those 197 calls” To what are you referring here? Were there 197 calls about non hormonal contraceptives?

- Is there a relationship between selected topics during IVR calls and change in knowledge/use of contraceptives? If so, the change in knowledge/behaviour could be addressed more clearly to the intervention. 

Discussion:

- The study does not directly show that the system leads to the change in outcome variables because there is no control group applied. Please adjust the wording. 

- Why has the use of implants increased so much compared to other contraceptives?

This could be an interesting topic for the discussion. 

- How do you explain the difference between most frequently used contraceptives (hormonal contraceptives, condoms) and most frequently selected IVR topics (natural contraceptives)?

6. PLOS authors have the option to publish the peer review history of their article (what does this mean?). If published, this will include your full peer review and any attached files.

**Do you want your identity to be public for this peer review?** For information about this choice, including consent withdrawal, please see our Privacy Policy.

Reviewer #1: Yes: Dr Garoma Kitesa Begna

Reviewer #2: No

Reviewer #3: No

---

## [Decision Letter · Decision Letter 1]

2 Feb 2023

PDIG-D-22-00061R1

"The phone number is telling us good things which we didn’t know before.” Use of Interactive Voice Response Calling for Improving knowledge and uptake of family planning methods among Maasai in Tanzania

PLOS Digital Health

Dear Dr. Ngowi,

Thank you for submitting your manuscript to PLOS Digital Health. After careful consideration, we feel that it has merit but does not fully meet PLOS Digital Health's publication criteria as it currently stands. Therefore, we invite you to submit a revised version of the manuscript that addresses the points raised during the review process.

EDITOR: Please insert comments here and delete this placeholder text when finished. Be sure to:

* Indicate which changes you require for acceptance versus which changes you recommend

* Address any conflicts between the reviews so that it's clear which advice the authors should follow

* Provide specific feedback from your evaluation of the manuscript

Please submit your revised manuscript within 30 days Mar 04 2023 11:59PM. If you will need more time than this to complete your revisions, please reply to this message or contact the journal office at digitalhealth@plos.org. Please include the following items when submitting your revised manuscript:

We look forward to receiving your revised manuscript.

Kind regards,

Yuan Lai, Ph.D.

Academic Editor

PLOS Digital Health

Journal Requirements:

Additional Editor Comments (if provided):

Reviewers' comments:

Reviewer's Responses to Questions

**Comments to the Author**

1. If the authors have adequately addressed your comments raised in a previous round of review and you feel that this manuscript is now acceptable for publication, you may indicate that here to bypass the “Comments to the Author” section, enter your conflict of interest statement in the “Confidential to Editor” section, and submit your "Accept" recommendation.

Reviewer #1: All comments have been addressed

Reviewer #3: (No Response)

2. Does this manuscript meet PLOS Digital Health’s publication criteria? Is the manuscript technically sound, and do the data support the conclusions? The manuscript must describe methodologically and ethically rigorous research with conclusions that are appropriately drawn based on the data presented.

Reviewer #1: Yes

Reviewer #3: Yes

3. Has the statistical analysis been performed appropriately and rigorously?

Reviewer #1: Yes

Reviewer #3: Yes

4. Have the authors made all data underlying the findings in their manuscript fully available (please refer to the Data Availability Statement at the start of the manuscript PDF file)?

Reviewer #1: Yes

Reviewer #3: Yes

5. Is the manuscript presented in an intelligible fashion and written in standard English?

Reviewer #1: Yes

Reviewer #3: Yes

6. Review Comments to the Author

Reviewer #1: they tried to incorporate the comments but still let they modify the English, other comments are just minor that could me modified and I just highlighted my concern in the document too. Thanks

Reviewer #3: Dear author,

considering the reviewers comments, your manuscript has improved substantially, congratulations on that. However, not all comments were implemented properly. There are still some minor issues.

General: 

- Formatting: The formatting of the text is still faulty in many places. There are double spaces, misplaced commas and dots, unfinished phrases (e.g. Abstract), and citations outside the phrase in the manuscript. Please review the manuscript carefully. 

Abstract:

- double use of "increased" in Results section

- last sentence in Results section incomplete

Methods:

- Second sentence grammatically wrong "This was serial design consisted of a baseline assessment ..."

- Please add the information that participants without a cellphone were provided with one to the manuscript.

Results:

- Page 15: The final quote on that page was already used on page 14, but with a different reference.

Discussion:

- Page 20 first sentence: that or which

Best wishes!

7. PLOS authors have the option to publish the peer review history of their article (what does this mean?). If published, this will include your full peer review and any attached files.

**Do you want your identity to be public for this peer review?** For information about this choice, including consent withdrawal, please see our Privacy Policy. 

Reviewer #1: Yes: Dr Garoma Kitesa Begna

Reviewer #3: No

---

## [Decision Letter · Decision Letter 2]

11 Apr 2023

“The phone number tells us good things we didn’t know before.” Use of Interactive Voice Response Calling for Improving knowledge and uptake of family planning methods among Maasai in Tanzania

PDIG-D-22-00061R2

Dear Mr Ngowi,

We are pleased to inform you that your manuscript '“The phone number tells us good things we didn’t know before.” Use of Interactive Voice Response Calling for Improving knowledge and uptake of family planning methods among Maasai in Tanzania' has been provisionally accepted for publication in PLOS Digital Health.

Best regards,

Yuan Lai, Ph.D.

Academic Editor

PLOS Digital Health

As reviewer 2 pointed out, some format and editing issues need to be fixed. Please address reviewer 2's comments before final proofreading and publication.

Reviewer Comments (if any, and for reference):

Reviewer's Responses to Questions

**Comments to the Author**

1. If the authors have adequately addressed your comments raised in a previous round of review and you feel that this manuscript is now acceptable for publication, you may indicate that here to bypass the “Comments to the Author” section, enter your conflict of interest statement in the “Confidential to Editor” section, and submit your "Accept" recommendation.

Reviewer #1: All comments have been addressed

Reviewer #3: (No Response)

2. Does this manuscript meet PLOS Digital Health’s publication criteria? Is the manuscript technically sound, and do the data support the conclusions? The manuscript must describe methodologically and ethically rigorous research with conclusions that are appropriately drawn based on the data presented.

Reviewer #1: Yes

Reviewer #3: (No Response)

3. Has the statistical analysis been performed appropriately and rigorously?

Reviewer #1: Yes

Reviewer #3: (No Response)

4. Have the authors made all data underlying the findings in their manuscript fully available (please refer to the Data Availability Statement at the start of the manuscript PDF file)?

Reviewer #1: Yes

Reviewer #3: (No Response)

5. Is the manuscript presented in an intelligible fashion and written in standard English?

Reviewer #1: Yes

Reviewer #3: (No Response)

6. Review Comments to the Author

Reviewer #1: I want to my appreciation to the team for their great effort....thanks

Reviewer #3: Still not addressed:

- Abstract: last sentence in Results section incomplete

- Page 15: The final quote on that page was already used on page 14, but with a different

reference.

- Please add the information that participants without a cellphone were provided with one

to the manuscript. (Still cannot find the information in the manuscript.

7. PLOS authors have the option to publish the peer review history of their article (what does this mean?). If published, this will include your full peer review and any attached files.

**Do you want your identity to be public for this peer review?** For information about this choice, including consent withdrawal, please see our Privacy Policy.

Reviewer #1: **Yes: **Dr Garoma Kitesa Begna

Reviewer #3: No
